# Endoplasmic Reticulum Stress and Its Impact on Adipogenesis: Molecular Mechanisms Implicated

**DOI:** 10.3390/nu15245082

**Published:** 2023-12-12

**Authors:** Gyuhui Kim, Jiyoon Lee, Joohun Ha, Insug Kang, Wonchae Choe

**Affiliations:** 1Department of Biomedical Science, Graduate School, Kyung Hee University, Seoul 02447, Republic of Korea; eieiclo@khu.ac.kr (G.K.); hajh@khu.ac.kr (J.H.); iskang@khu.ac.kr (I.K.); 2Department of Biochemistry and Molecular Biology, School of Medicine, Kyung Hee University, Seoul 02447, Republic of Korea; 3Department of Biological Sciences, Franklin College of Arts and Sciences, University of Georgia, Athens, GA 30609, USA; jl86599@uga.edu

**Keywords:** endoplasmic reticulum stress, adipogenesis, unfolded protein response, obesity, metabolic disorders

## Abstract

Endoplasmic reticulum (ER) stress plays a pivotal role in adipogenesis, which encompasses the differentiation of adipocytes and lipid accumulation. Sustained ER stress has the potential to disrupt the signaling of the unfolded protein response (UPR), thereby influencing adipogenesis. This comprehensive review illuminates the molecular mechanisms that underpin the interplay between ER stress and adipogenesis. We delve into the dysregulation of UPR pathways, namely, IRE1-XBP1, PERK and ATF6 in relation to adipocyte differentiation, lipid metabolism, and tissue inflammation. Moreover, we scrutinize how ER stress impacts key adipogenic transcription factors such as proliferator-activated receptor γ (PPARγ) and CCAAT-enhancer-binding proteins (C/EBPs) along with their interaction with other signaling pathways. The cellular ramifications include alterations in lipid metabolism, dysregulation of adipokines, and aged adipose tissue inflammation. We also discuss the potential roles the molecular chaperones cyclophilin A and cyclophilin B play in adipogenesis. By shedding light on the intricate relationship between ER stress and adipogenesis, this review paves the way for devising innovative therapeutic interventions.

## 1. Introduction

The prevalence of obesity and associated metabolic disorders is on the rise as a result of sedentary lifestyles and high caloric intake [1]. Obesity results from genetic, environmental, and lifestyle factors, and excessive endoplasmic reticulum (ER) stress is believed to play a role [2,3,4]. The ER is involved in folding and assembling proteins while facilitating the movement of newly synthesized proteins to their required destinations, lipid synthesis, and regulation of calcium levels [5]. ER stress occurs when there is an influx of proteins into the ER beyond its processing capacity, reaching a limit in protein folding, or depletion of calcium within the ER [6]. Prolonged ER stress leads to cellular apoptosis. To prevent this, the ER regulates homeostasis through the unfolded protein response (UPR) [7].

The UPR responds to ER stress by reducing protein translation [8,9], upregulating chaperones, promoting folding [9], and degrading misfolded proteins [10,11,12]. The inositol-requiring enzyme1—X-box binding protein 1 (IRE1-XBP1), protein kinase RNA-like ER kinase (PERK) and activating transcription factor 6 (ATF6) pathways impact both ER stress and adipogenesis, influencing adipogenic transcription factors such as peroxisome proliferator-activated receptor γ (PPARγ) and CCAAT-enhancer-binding proteins (C/EBPs) [13].

The escalation of ER stress can perpetuate obesity in a vicious cycle [4]. Elevated ER stress has the potential to stimulate inflammatory responses associated with adipocytes and insulin resistance [14]. This stimulation may lead to increased energy storage and exacerbation of obesity, consequently further amplifying ER stress [3,15,16]. These research findings underscore the bidirectional relationship between ER stress and obesity, highlighting that their interaction is not merely a unidirectional cause-and-effect scenario but rather manifests as a mutually reinforcing connection. Understanding ER stress-adipogenesis interplay is crucial for combating obesity, since investigating the relationship between ER stress and adipogenesis provides valuable insights into the origins of obesity and potential treatments. Therefore, understanding ER stress-induced adipogenesis could lead to innovative strategies for combating obesity and its associated complications. Also, targeting ER stress could alleviate adipose tissue dysfunction and metabolic issues.

This review explores the dysregulation of UPR pathway, ER stress’s impact on adipogenesis, and therapeutic interventions for obesity-related metabolic disorders.

## 2. The Process of Adipogenesis

Adipogenesis is the formation process of adipocytes which entails cell conversion into adipose tissue followed by accumulating lipids within the cells and differentiation into adipocytes. In human physiology and health, adipogenesis holds profound significance since it involves the differentiation of precursor cells into mature adipocytes. These specialized fat cells serve as important energy storage reservoirs, releasing triglycerides during periods of heightened energy demand [17,18,19]. They also act as an insulating layer, regulating body temperature [20,21], and provide crucial protection to internal organs [22,23]. Additionally, fat cells act as endocrine cells, secreting hormones and adipokines such as leptin, adiponectin, and resistin, which play vital roles in regulating metabolism, appetite, and inflammatory processes [24,25,26,27,28]. Dysregulation of adipogenesis contributes to obesity, a significant risk factor for metabolic disorders such as type 2 diabetes, cardiovascular diseases, and fatty liver disease [29,30,31,32]. Therefore, comprehending this complex biological process is imperative for advancing human health.

Adipogenesis involves the formation and differentiation of fat cells or adipocytes, progressing through several critical stages. In theory, the process starts with multipotent mesenchymal stem cells (MSCs), capable of differentiating into various cell types, including adipocytes [33]. These cells commit to becoming adipocytes along one of the numerous differentiation pathways available [34].

As differentiation begins, crucial genes such as C/EBPs and PPARγ are activated [35,36]. These proteins have a vital role in the initial stages of adipocyte differentiation. As shown in Figure 1, the expression of C/EBPs and PPARγ starts the conversion of MSCs into adipocytes, allowing these cells to obtain the ability to form fat and transform into adipose tissue [37,38].

Differentiated adipocytes initiate the synthesis of triacylglycerol (TAG), a major neutral fat stored within fat cells [39,40]. Consequently, the fat content inside the cell increases, culminating in the formation of fully mature adipocytes [41]. These adipocytes typically exhibit a larger, round or spherical shape and contain substantial lipid droplets, serving as critical components of adipose tissue [42,43].

Adipogenesis plays a critical role in energy storage and metabolic regulation, rendering it a prominent research area, notably concerning obesity and metabolic disorders. This process leads to the formation of adipose tissue, which contributes to energy balance and metabolic regulation.

## 3. UPR Signaling Pathways (IRE1-XBP1, PERK, ATF6) in Obesity

The UPR is conducted by three primary transmembrane proteins present on the ER membrane: IRE1, PERK, and ATF6 [44]. These sensors can detect ER stress and activate adaptive signaling pathways aimed at restoring ER homeostasis [45].

The IRE1 pathway involves the non-traditional splicing of XBP1 mRNA, which produces a spliced form called XBP1s [46]. This transcription factor encourages the expression of chaperones, foldases, and ER-associated degradation (ERAD) pathway constituents [47,48,49]. This leads to improved folding capacity of the ER [50]. The PERK pathway leads to the phosphorylation of eukaryotic translation initiation factor 2 alpha (eIF2α), which reduces overall protein translation [51]. Despite this decrease, it eases the ER protein load and triggers the translation of ATF4 [52]. This promotes the expression of genes related to antioxidant responses and amino acid metabolism [53]. In contrast, the ATF6 pathway involves ATF6 translocating to the Golgi apparatus under ER stress [47]. The cleavage of ATF6 results in a transcriptionally active fragment that enters the nucleus, improving the ER’s ability to manage protein folding demands by enhancing the expression of chaperones and ERAD-related factors [54].

The UPR acts as a balance between adaptive mechanisms that mend ER homeostasis and a terminal response that triggers apoptosis in highly stressed cells [55,56,57]. ER homeostasis is maintained by ER chaperone proteins such as glucose-regulated protein 78 (GRP78), GRP94, calreticulin (CRT), and protein disulfide isomerase (PDI) [58,59,60,61,62]. Particularly, GRP78 (also known as immunoglobulin heavy chain-binding protein or BiP) is a well-characterized member of the heat shock protein 70 kDa (HSP70) family, encoded by the HSPA5 gene, which is essential for proper protein folding, regulation of the UPR signaling, maintaining chaperone balance, and preventing apoptosis [63,64,65,66]. Among its crucial roles, GRP78 facilitates proper protein folding within the ER, maintains proteins in their folded state, prevents aggregation of protein folding intermediates, and directs misfolded proteins to the ERAD pathway [67,68,69].

Additionally, GRP78 plays a significant role in maintaining intracellular calcium (Ca^2+^) homeostasis within the ER [70]. It regulates intracellular Ca^2+^ levels and contributes to various cellular processes involving Ca^2+^ signaling [71]. Furthermore, under specific cellular stress conditions, GRP78 can form complexes with pro-caspases such as caspase-7 and caspase-12 on the ER membrane, providing protective functions for cell survival [72,73].

Through these diverse functions, GRP78 plays a pivotal role in balancing cell survival and apoptosis in cells experiencing ER stress [66,73]. It is also necessary during early embryonic development and exhibits reduced expression during aging [65,74].

Obesity, which is defined as the excessive accumulation of adipose tissue, is linked to a chronic condition of low-grade inflammation and metabolic disorders [4,75,76]. When there is an abundance of nutrients, adipocytes face a challenge to produce and secrete significant amounts of adipokines and cytokines, increasing the risk of ER stress [2,77,78]. To cope with the heightened demand for protein folding, the UPR is activated to restore ER function [79]. This impacts adipocyte function significantly. The activation of the UPR in adipocytes has a dual influence on obesity. Firstly, it strives to restore ER homeostasis by boosting the expression of chaperone proteins and increasing the ER folding capacity [80]. Secondly, persistent ER stress can disrupt UPR signaling, resulting in cellular dysfunction and insulin resistance [30].

### 3.1. The IRE1-XBP1 Pathway and Lipid Metabolism

The IRE1-XBP1 pathway, a pivotal component of the UPR, is connected with lipid metabolism and adipogenesis [81,82]. Via IRE1, the splicing of XBP1 has a direct impact on the expression of lipogenic genes and lipid droplet dynamics [5]. In such scenarios, IRE1 becomes activated and splices XBP1 mRNA [83]. XBP1 mRNA splicing converts the inactive XBP1 (XBP1u) into its active form, XBP1s, through IRE1’s ribonuclease activity [84]. XBP1s, once activated, functions as a transcription factor that regulates the expression of different genes. Specifically, XBP1s promotes the expression of ERAD genes which facilitate the elimination of unnecessary misfolded proteins [85]. When specifically deleting the XBP1 gene in the adult mouse liver to investigate its function, we observed a significant reduction of approximately 85–90% in hepatic fatty acid and cholesterol synthesis. This led to lowered concentrations of plasma cholesterol and triglycerides [86]. Furthermore, hepatic overexpression of XBP1 directly upregulates the promoters of lipid synthesis genes, including acetyl-CoA carboxylase 2 (ACC2) and sterol regulatory element-binding Protein 1 (SREBP1), thereby promoting lipid synthesis [87]. In mouse hepatic cells, XBP1 exacerbates lipid synthesis and suppresses lipid breakdown, thereby worsening lipid accumulation. However, XBP1 knockout in mice reduces hepatic steatosis, increases lipid breakdown, and decreases lipid accumulation. Consequently, pharmacologically inhibiting XBP1 presents a new potential for treating non-alcoholic fatty liver disease (NAFLD) [88].

XBP1s induces the expression of various genes, such as FAS, SREBP1c, ACC, DGAT, ChREBP, PLIN, CIDE, ATGL, HSL, and others, as shown in Table 1 [47,86,89,90,91]. This regulation allows XBP1s to exert a significant impact on physiological processes related to lipid metabolism and obesity.

Collectively, the IRE1-XBP1 pathway detects ER stress and responds by regulating lipid metabolism. XBP1s functions as a transcription factor, overseeing the expression of diverse genes, thereby aiding in lipid generation, storage, and regulation. Disrupting this pathway due to lengthy ER stress can lead to abnormal lipid accumulation and malfunctioning adipocytes, thus promoting obesity-related ailments.

### 3.2. The PERK Pathway and Insulin Sensitivity

PERK regulates protein synthesis to oversee the correct folding of proteins within the ER and the accumulation of defective proteins [92]. To overcome imbalances caused by ER stress, PERK promotes the phosphorylation of eIF2α, leading to the temporary inhibition of protein synthesis [93]. Consequently, cells can withstand stress, ensuring survival through appropriately regulated protein synthesis [94].

PERK exists in a homomeric form under stable conditions but transitions into a tetrameric structure under stress conditions, leading to trans-autophosphorylation of the PERK domain at the C-terminus [95,96].

Pancreatic islet β cells are specialized secretory cells responsible for insulin storage, and they produce more insulin in insulin-resistant states [97,98]. In this context, processes such as proinsulin folding, ERAD, and mediation of quality/quantity control, as well as trafficking, are regulated to manage metabolic states and insulin demand [99,100,101,102].

Furthermore, the PERK-ATF4 pathway plays a crucial role in β cell biology and diabetes research [103,104]. PERK deficiency induces ER stress and high blood glucose levels, and PERK-mediated phosphorylation of eIFα is associated with glucose intolerance [105,106,107]. However, reduced PERK activity promotes glucose-stimulated insulin secretion (GSIS), and deletion of downstream signaling factors in the PERK-ATF4 pathway helps alleviate ER stress and prevent β cell loss [108,109,110].

Moreover, in the absence of PERK, the activity of enzymes involved in lipid production such as SREBP-1c, FAS, and SCD1 is hindered, and PERK accumulates in lipid droplets [111]. Additionally, during the differentiation process of fat cells, PERK has demonstrated its utilization of diacylglycerol to activate lipid kinases [112]. Research confirms that the downregulation of PERK reduces adipogenesis by decreasing ATF4 [13]. ATF4 has demonstrated active regulation of adipocyte differentiation across various evidence. Overexpression of ATF4 in 3T3-L1 cells enhances adipogenesis, while ATF4 siRNA inhibits pre-adipocyte differentiation into mature adipocytes. Depletion of ATF4 reduces adipocyte differentiation in human mesenchymal stem cells [113]. Recent studies observed elevated phosphorylation of PERK, an ER stress marker, in obese mice on a high-fat diet (HFD). This heightened phosphorylation, compared with normal diet-fed mice, correlates with abnormal protein degradation and increased lipid accumulation [114]. PERK utilizes its intrinsic lipid kinase activity to generate phosphatides, mediating Akt activation, thereby promoting adipocyte differentiation [112,115]. Consequently, PERK can stimulate adipocyte differentiation through Akt activation [116]. Additionally, ATF6α pathway activation also contributes to adipogenesis [117].

Thus, pathways associated with PERK significantly influence insulin sensitivity and β cell function, playing a crucial role in diabetes research and obesity.

### 3.3. The ATF6 Pathway and Inflammation

ATF6 is a transmembrane transcription factor with an N-terminal domain in the cytoplasm and a C-terminal domain in the ER lumen [118]. ATF6 contributes towards ERAD for resolving incorrect protein folding [54]. In mammalian cells, ATF4 and ATF6 are reported to interact with the 26S proteasome, inducing the ER membrane protein HERP/Mif1 and facilitating efficient ERAD [119]. This arrangement positions the proteasome closer to the ER, enabling smoother protein degradation [44]. The UPR is suggested to function in two stages. In the first stage, it allows time for protein folding, and in the second stage, it targets unfolded proteins for degradation [120]. ATF6’s rapid activation is compared with IRE1, which is believed to occur due to ATF6’s swift activation compared with IRE1, responsible for inducing XBP1 splicing and translation. During this period, ATF6-induced ER chaperones can facilitate protein folding before inducing ERAD genes that promote the degradation of unfolded XBP1 [120,121]. 

ATF6, initially located on the ER membrane, moves to the Golgi apparatus under ER stress [47], where it is cleaved by site 1 protease (S1P) and site 2 protease (S2P) to form the N-terminal fragment [122,123]. The N-fragment translocates to the nucleus and serves as a transcription factor [124]. One study confirmed that protein kinase B (AKT) phosphorylation mediated by ATF6 contributes to downstream nuclear factor-kappa B (NF-κB) activation [125,126,127]. This interaction assists NF-κB, which regulates inflammation and immune responses, by inducing the expression of inflammatory genes. The association of ATF6 with NF-κB upregulates the expression of inflammation-associated genes, including the cytokines interleukin-6 (IL-6) and tumor necrosis factor-alpha (TNF-α) [128]. Furthermore, ATF6 can induce cell death during prolonged ER stress by activating downstream effectors, including CHOP, c-Jun N-terminal kinase (JNK), and proapoptotic Bcl-2 family proteins [129]. Dysfunctional signaling of ATF6 may contribute to the accumulation of proteins that are misfolded and exacerbate inflammation, which is a characteristic of obesity [2]. 

Inhibition of ATF6 in mesenchymal stromal C3H10T1/2 cells impedes lipid accumulation, downregulating crucial genes for adipogenesis: PPARγ, SREBP-1c, GLUT4, and aP2 [117]. PPARγ reduction intensifies during adipogenesis in ATF6-deficient cells versus controls [5]. Diminished ATF6 correlates with restrained C/EBPβ, an early adipogenic factor. Although direct regulation is not confirmed, ATF6 overexpression increases acetyl-CoA carboxylase beta (Acacb) and Fasn expression in mouse embryonic fibroblasts (MEF) and enhances FAS in Chinese hamster ovary (CHO) cells [130]. The ER stress pathways (IRE1, PERK, ATF6) collectively drive both lipogenesis and adipogenesis [5,131]. Inhibiting all of them is pivotal to curb lipogenesis and delay the onset of obesity.

## 4. ER Stress and Adipogenesis

### 4.1. Effects of ER Stress on Transcription Factor Involed in Adipogenesis (PPARγ, C/EBPs)

The impact of ER stress on transcription factors that control fat, specifically peroxisome PPARγ and C/EBPs, has recently gained significance in research. PPARγ and C/EBPs are crucial in regulating adipocyte differentiation and lipid metabolism [132]. Under normal physiological conditions, these transcription factors coordinate the expression of genes involved in adipogenesis, adipocyte maturation, and lipid storage [36]. However, during ER stress, the phosphorylation of eIF2α in the PERK-eIF2α pathway also increases the translation of C/EBP in an in vitro model [133]. Similarly, reduced phosphorylation of eIF2α achieved by overexpression of GADD34 in the liver decreases the expression of C/EBPα, C/EBPβ, and PPARγ [134].

The typical adipogenesis progression in 3T3-L1 preadipocytes involves three stages: first, they undergo contact inhibition, then mitotic clonal expansion (McE), followed by the final stage of adipogenic differentiation [135]. Initially, preadipocytes express adipogenic transcription factors such as CCAAT-enhancer-binding proteinβ/δ (C/EBPβ/δ) and exhibit low levels of PPARγ [136]. Interestingly, early induced C/EBPβ is inactive in preadipocytes, while PPARγ serves as a crucial master regulator in the transcriptional program of adipocytes [137]. C/EBPβ and C/EBPδ are early inducers of adipocyte differentiation and promote the expression of CCAAT-enhancer-binding Protein α (C/EBPα) and PPARγ, key regulators of mature adipocyte function [36].

C/EBPα is required for the accumulation of lipids and insulin sensitivity in differentiated adipocytes [138]. The transition from preadipocytes to mature adipocytes is initiated by pro-adipogenic signals including insulin, dexamethasone, 3-isobutyl-1-methylxanthine (IBMX), or bone morphogenetic proteins (BMPs). This process entails increasing the expression of adipogenic transcription factors, such as PPARγ and C/EBPα, resulting in morphological alterations from fibroblast-like cells to spherical ones with a solitary prominent lipid droplet [36,139]. Over time, mature adipocytes demonstrate metabolic and endocrine characteristics, supported by genes such as fatty acid-binding protein 4 (FABP4), glucose transporter type 4 (GLUT4), leptin, and adiponectin [140,141].

PPARγ plays a key role in adipocyte differentiation, with two isoforms referred to as PPARγ1 and PPARγ2 [142,143,144]. Both isoforms promote adipocyte differentiation, although PPARγ2 demonstrates more efficiency at lower ligand concentrations [144,145]. Additionally, C/EBPs, a group of transcription factors, are crucial for adipogenesis [132]. They stimulate the production of C/EBPα, which is vital for insulin sensitivity in differentiated adipocytes [146]. The complex molecules involved in adipogenesis propose that mitotic clonal expansion could produce internal ligands for PPARγ [147,148]. Additional study is necessary to completely clarify the complicated connection between peroxisome proliferator-activated receptors (PPARs), C/EBPs, and ER stress in the context of adipogenesis.

Understanding the complex relationship between ER stress and fat transcription factors, namely, PPARγ and C/EBPs, is essential to uncovering the molecular mechanisms that drive adipogenesis and metabolic dysregulation. By elucidating the effects of ER stress on these transcription factors, new therapeutic approaches targeting ER stress reduction and proper function restoration may be developed, providing potential interventions for obesity and related metabolic disorders.

### 4.2. Relationship between CHOP and a Transcription Factor Involved in Adipogenesis

ER stress induces the expression of interleukin-8 (IL-8), an inflammatory cytokine, and the nuclear translocation of CHOP [149]. This upregulation of IL-8 due to ER stress subsequently leads to an increase NF-κB expression [150,151]. NF-κB is a factor that is negatively regulated by the adipogenic differentiation factor PPARγ [149,152]. The activity of PPARγ serves as a crucial regulator in maintaining balance by inhibiting NF-κB and decreasing inflammatory responses [153,154].

ER stress-induced CHOP expression is induced through the UPR, typically through the PERK pathway [151]. CHOP is a transcriptional regulator within the nucleus and regulates numerous genes involved in cellular processes such as inflammation, differentiation, autism, and apoptosis [155,156,157,158]. CHOP is a stress response element that responds to cellular insults such as ER stress and nutrient deprivation and is dependent on eIF2α phosphorylation [159]. It also plays a role in various inflammatory responses [160]. Moreover, endotoxemia enhances CHOP activity, which leads to caspase processing of interleukin-1β (IL-1β) [161].

CHOP impedes the differentiation of the mesenchymal lineage [162]. It is a crucial regulator of adipogenesis, and this function is supported by various experiments. CHOP is recognized to be a principal inhibitory factor for the adipogenic differentiation factor C/EBPβ, and it can hinder the downstream targets of C/EBPβ, such as PPARγ [154,162]. CHOP has a negative impact on the initial stages of adipogenic differentiation by inhibiting the activation of C/EBPβ, which then affects the activation of C/EBPα and PPARγ [163]. The inhibition of CHOP also fortifies the binding of C/EBPα to PPARγ and increases PPARγ promoter activity in response to intracellular ER stress [149]. For effective fat storage, the final differentiation of adipocytes is necessary [164]. CHOP was first discovered to inhibit the differentiation of adipocytes in response to metabolic stress, hypoxia, and phosphorylation induced by p38 MAPK [159,165]. Subsequent activation of PERK-eIF2α during ER stress results in suppressed biphasic differentiation through CHOP expression [53]. Under conditions of polyamine depletion, CHOP interacts with C/EBPβ to inhibit the execution of the tin dioxide clonal expansion process and transcriptional activation of adipogenesis. This results in an inhibitory effect [163]. Overexpression of CHOP leads to poorly differentiated adipocytes and an increase in undifferentiated adipose tissue in a mouse model [166]. Inhibition of CHOP mRNA is required for full adipocyte differentiation of MEFs [159].

### 4.3. Crosstalk between ER Stress and Other Signaling Pathways in Adipogenesis

Adipogenesis involves adipocyte differentiation and maturation regulated by a network of signaling pathways [167]. Recent evidence indicates that ER stress, defined as the buildup of misfolded proteins in the ER, collaborates with other signaling pathways to regulate adipogenesis [168,169,170,171]. This section seeks to explore the complex molecular interactions and crosstalk between ER stress and other signaling pathways implicated in adipogenesis.

#### 4.3.1. ER Stress and UPR in Adipogenesis

In an obese environment, fat accumulation within cells can lead to protein folding issues during processes such as fatty acid synthesis in adipocytes [43]. These problems result in a larger number of incomplete protein folds in the ER and the activation of the UPR [4]. ER stress, a component of the UPR, can restrict protein synthesis through signaling pathways, which inhibits adipocyte differentiation and consequently limits adipocyte formation [111]. During ER stress, cells use eIF2α as a protein guide [166]. eIF2α plays a crucial part in the initial stages of protein synthesis and undergoes regulation in a unique way under ER stress conditions [172]. Phosphorylation of eIF2α occurs via guidepost proteins due to ER stress [173]. This blocks protein synthesis by hindering the communication with eIF2B, which facilitates the transfer of methionine from nucleic acid tRNA to the ribosome during the initial phases of protein synthesis [174].

When these mechanisms operate in unison to trigger ER stress, this represses protein synthesis, ultimately hindering the differentiation of adipocytes [175,176,177].

#### 4.3.2. ER Stress and Wingless/Integrated (Wnt) Signaling in Adipogenesis

Wnt signaling is a crucial pathway in adipogenesis, governing the determination of preadipocyte fate and adipocyte maturation [178,179]. Recent research highlights the possibility of crosstalk between ER stress and Wnt signaling, implying the ability of ER stress to regulate the key Wnt signaling components [180,181]. Wnt signaling occurs when Wnt proteins bind to frizzled receptors [182,183,184]. This activates signaling pathways that are both dependent and independent of β-catenin [185]. Importantly, Wnt signaling represses adipocyte differentiation through suppression of adipogenic transcription factors, such as PPARγ and C/EBPα [181]. Wnt10b exhibits constitutive expression, mainly in preadipocytes and stromal vascular cells—not adipocytes—and significantly impedes adipogenesis [179,181,186]. In vivo, transgenic expression of Wnt10b in adipocytes leads to a 50% decrease in total body fat and lack of brown adipose tissue formation, emphasizing the intricate nature of Wnt signaling in adipogenesis [186]. The interplay between Wnt signaling and adipogenesis implies that preadipocytes integrate signals from numerous Wnt pathways, which ultimately influences the expression of vital adipogenic regulators such as PPARγ and C/EBPα, affecting adipocyte differentiation and development.

#### 4.3.3. ER Stress and mTOR Signaling in Adipogenesis

Mammalian target of rapamycin (mTOR) is a significant protein that regulates both cell growth and metabolism, playing a crucial role in adipocyte differentiation and lipid metabolism [187,188]. Recent research has indicated that ER stress modulates mTOR signaling. Moreover, the activation of ER stress can hinder the activation of mTOR complex 1 (mTORC1) [189,190].

The mechanism involved in the interaction between ER stress and mTORC1 is intriguing. Notably, ER stress is associated with AMP-activated protein kinase (AMPK) [191,192]. ER stress triggers a pathway that reduces cellular ATP levels and increases AMP levels, resulting in the activation of AMPK [193]. AMPK senses and regulates cellular energy status [194]. Its activation leads to the inhibition of mTORC1 by activating tuberous sclerosis complex1-tuberous sclerosis complex 2 (TSC1-TSC2), which inhibits Rheb protein required for mTORC1 activation [188,190].

Furthermore, AMPK phosphorylates raptor, one of mTORC1’s subunits, leading to the inhibition of raptor’s activity and control over mTORC1 activation [195,196]. The AKT-mTORC1 pathway regulates lipid synthesis via the sterol regulatory element-binding protein (SREBP) transcription factor [187,197]. When ER stress is triggered, AKT’s activation is impeded [198].

ER stress triggers ATF4 translation, which fosters cellular apoptosis by means of inhibiting AKT through stress-related proteins, such as TRB3 and others [199]. Moreover, ER stress obstructs mTORC2 and AKT via the GSK-3β pathway, leading to the activation of the IRE1-JNK pathway and ultimately inducing cell apoptosis [131,200,201].

In summary, AKT and AMPK function as significant signaling nodes pertaining to the activation and inhibition of mTORC1 [190,200]. Additionally, ER stress plays a regulatory role in these interactions [191].

#### 4.3.4. ER Stress and Insulin Signaling in Adipogenesis

Insulin signaling pathways are closely connected to adipogenesis and metabolic homeostasis [187]. Experimental models [202] confirm ER stress’s role in obesity-related insulin resistance. Increased ER stress has been linked to impaired insulin action in obese mice [203], and chemical or genetic modification of this stress has been shown to improve insulin sensitivity and glucose homeostasis [202]. In cases where tissues, such as liver, skeletal muscle, and fat, become less responsive to insulin, signal transmission is reduced for insulin receptor substrate (IRS) [204], AKT, and glycogen synthase kinase-3β (GSK3β) [205,206]. Previous studies suggest that increased levels of interleukin-6 (IL-6) and TNF-α may be linked to obesity and insulin resistance [207], signifying their involvement in ER stress and reduced insulin sensitivity [208,209]. These results indicate a considerable role for cytokines [210]. Research findings show that ER stress can interfere with insulin signaling pathways by activating serine kinases, including JNK and inhibitor of nuclear factor kappa-B kinase (IKK) [211,212,213]. Impairment of insulin signaling by ER stress can hinder adipocyte differentiation, ultimately contributing to insulin resistance [214].

IRE1 signaling pathway-induced activation of JNK and subsequent activation of inflammatory signaling pathways are pivotal factors in the development of insulin resistance and type 2 diabetes (T2DM) associated with obesity [211,215,216]. A number of studies have emphasized this process as a crucial component of the pathophysiology of insulin resistance related to obesity [217].

ER stress caused by obesity stimulates JNK activation, which acts as a core mediator resulting in modifications in insulin signaling [218,219]. The activation of JNK is primarily responsible for the changes in insulin signaling pathways that contribute to insulin resistance [211].

ER stress and inflammation in obesity result in the elevation of pro-inflammatory cytokines, including IL-6 and TNF-α [4,220]. These raised cytokine levels impair insulin action and promote insulin resistance [220,221].

When cells encounter pro-inflammatory cytokines or high levels of free fatty acids (FFA), they hinder insulin signaling by phosphorylating serine residues on the insulin receptor substrate-1 (IRS-1) [222]. This phosphorylation disrupts insulin signaling downstream and impairs insulin function [220].

Moreover, JNK is activated and phosphorylates IRS-1 when cells encounter stimuli such as ER stress, elevated cytokine levels, or high levels of fatty acids [131,219]. IRS-1 and insulin receptor substrate-2 (IRS-2) play a vital role as substrates for the insulin receptor tyrosine kinase in the insulin signaling pathway [223,224]. This action, in turn, reduces the receptor’s sensitivity to insulin [223]. Consequently, overexpression of inflammatory molecules that result in the removal of IRS-1/2 receptors impede the insulin signaling pathway and lead to insulin resistance [219].

In summary, the activation of JNK by IRE1 disrupts the signaling of the insulin receptor, leading to insulin resistance [225]. This process is facilitated by ER stress and inflammatory cytokines, both playing crucial roles [221].

#### 4.3.5. ER Stress and Nuclear Receptors in Adipogenesis

The nuclear receptors, specifically the PPARs and the liver X receptors (LXRs), assume critical roles in adipogenesis and lipid metabolism [226]. These molecular entities are responsible for regulating lipid metabolism and the development of adipocytes [227]. Notably, the LXRs belong to the class of nuclear receptors that exert a significant impact on both cholesterol metabolism and fat metabolism [228]. When ER stress inhibits LXRs, it may affect cholesterol and fat metabolism processes, resulting in increased cholesterol levels and abnormal fat metabolism [229].

ER stress-induced LXR inhibition negatively impacts cholesterol and lipid metabolic processes [230]. Activated LXRs trigger cholesterol metabolism genes, producing high-density lipoprotein (HDL) particles to contain cholesterol [231]. Furthermore, LXRs play a role in fat metabolism, regulating both oxidation and storage in adipose tissue [232]. Hence, impeding LXR regulation during ER stress can potentially cause abnormalities in fat metabolism, leading to abnormal fat accumulation [233]. Some significant genes involved in cholesterol metabolism are the ATP-binding cassette A1 (ABCA1) gene, which increases when LXRs are activated, and the ATP-binding cassette G1 (ABCG1) gene, which is also regulated by LXRs [234,235]. The ABCA1 gene is responsible for shuttling cholesterol and phospholipids and facilitates HDL particle production; its activation helps move cholesterol from cells to HDL [236]. Additionally, ABCG1 contributes to cholesterol transport. It aids in the transportation of fat phosphate, which is a constituent of HDL particles [237]. Additionally, LXRs regulate the gene expression of cholesteryl ester transfer protein (CETP), which is accountable for the transfer of cholesterol from HDL to other lipid particles [238].

Furthermore, the activation of LXR leads to an increase in lysophosphatidylcholine acyltransferase 3 (Lpcat3) expression [239]. This indicates that LXR identifies polyunsaturated fatty acids that encourage their absorption into phospholipids (PLs), thereby enhancing ER stability [239,240]. The link between Lpcat3 and LXR indicates that LXR activation escalates Lpcat3 expression, promoting the release of polyunsaturated PLs, subsequently contributing to higher ER membrane stability and minimizing ER stress [240]. This process of membrane remodeling decreases the stress on the endothelial membrane caused by saturated fatty acids [241]. Additionally, the LXR–Lpcat3 pathway mitigates hepatitis by regulating the activation of c-Src kinase and controlling the availability of lipid inflammatory mediators [242]. These findings underscore the significance of Lpcat3 regulation for regulating lipid balance in physiology and disease through LXR signaling [243,244].

This interaction is part of the intricate networks involved in adipocyte development [244]. The interaction between ER stress and nuclear receptors is crucial in comprehending and treating metabolic diseases including obesity, diabetes, and non-alcoholic fatty liver disease (NAFLD) [245].

Understanding the interplay between ER stress and other signaling pathways during adipogenesis yields key insights into the molecular mechanisms governing adipocyte development and function [246]. Disrupting these interactions may contribute to metabolic disorders and dysfunction of adipose tissue [247]. Further research is required to elucidate the precise molecular mechanisms relating to the interplay between ER stress and signaling pathways, as well as their potential implications for adipogenesis and metabolic health [248].

## 5. Cyclophilin Family in Adipogenesis

Cyclophilin A (CypA) and cyclophilin B (CypB) are both members of the cyclophilin protein family. They are peptidyl-prolyl cis-trans isomerases (PPIases) that catalyze peptidyl-prolyl bond isomerization in proteins. Despite their similar functions, they have distinct roles and cellular localizations [249].

CypA is a highly abundant, ubiquitously cytosolic protein present in various cell types and tissues [212]. It is primarily recognized for its function in mediating the immunosuppressive effects of the immunosuppressive medication cyclosporine A (CsA) [250]. CypA plays a pivotal role in the immune response by binding with the protein calcineurin and inhibiting its phosphatase activity, ultimately blocking T-cell activation and the production of pro-inflammatory cytokines [251,252,253].

CypB is predominantly located within the lumen of the ER [254]. Its main roles involve protein folding and ERAD [255,256]. It functions as a molecular chaperone by aiding in the correct folding of newly produced proteins in the ER and supporting their transportation to their intended destinations [257]. Additionally, CypB is involved in numerous cellular processes such as collagen biosynthesis and virus replication [258,259].

Although both CypA and CypB are peptidyl-prolyl isomerases and share some functional similarities, they have distinct roles in different cellular compartments [249]. CypA is primarily involved in immune regulation, while CypB functions in protein folding and quality control within the ER [251].

### 5.1. CypA

Recent research has confirmed that CypA is a critical regulator of fat production [260,261]. According to the study’s findings, CypA has emerged as a key factor in fat metabolism and its association with obesity. Experimental results have shown that CypA promotes fat production in test tubes and plays a role in contributing to obesity induced by a high-fat diet (HFD) in mice. CypA was also found to be associated with offspring obesity induced by maternal gestational diabetes in mice [260]. The 3T3L1 cells used in the study are progenitor cells that differentiate into adipocytes upon insulin stimulation [262]. An increase in CypA expression was observed on day 6 of the 8-day process of these cells differentiating into adipocytes. It has been reported that insulin affects adipocyte differentiation by regulating the expression of key transcription factors involved in adipogenesis, including CypA and PPARγ, C/EBPα and C/EBPβ. Specifically, silencing or knocking out CypA significantly reduced the expression of C/EBPβ in the early stages of adipocyte differentiation and reduced the expression of PPARγ, C/EBPα and C/EBPβ in the late stages of differentiation.

However, other studies suggest that CypA-CD147 interaction mediates obesity-induced macrophage–adipocyte crosstalk and, thus, may represent a novel target for the treatment of insulin resistance and type 2 diabetes [263]. CypA activates the surface receptor CD147, thereby activating NF-κB signaling, which increases the expression of pro-inflammatory cytokines, inducing adipocyte inflammation. Simultaneously, it hinders adipocyte differentiation by suppressing the expression of PPARγ and C/EBPβ through leucine zipper tumor suppressor 2 (LZTS2) mediated downregulation of β-catenin [263,264]. These findings suggest that CypA may attenuate adipose tissue function and improve insulin sensitivity. However, this study still leaves some unknowns that require further investigation.

In conclusion, these studies shed light on the effects of CypA on fat production and metabolism, suggesting its potential importance in obesity-related diseases. However, further research is needed to clarify the exact roles of these proteins and the underlying mechanisms, particularly the direct interaction mechanism between CypA and ER stress-related proteins, which remains to be elucidated.

### 5.2. CypB

CypB functions as a molecular chaperone predominantly located in the lumen of the ER, facilitating protein folding through its PPIase activity [256,265]. CypB is known to be related to cellular collagen formation and the growth of various cancer cells [266,267]. However CypB’s influence extends significantly into the intricate realm of adipogenesis, where it interacts with regulatory factors [268]. Recent evidence highlights the transcriptional upregulation of CypB as a response to ER stress. Interestingly, increased CypB expression triggers an enhanced interaction between CHOP and p300, an ER-resident proteasome [269]. This interaction, in turn, initiates the ubiquitination-driven degradation of CHOP, culminating in the attenuation of apoptotic effects during ER stress [269]. Furthermore, CypB reveals an intriguing facet of its functionality in the inflammatory setting, as it intricately modulates ER calcium levels and counteracts the accumulation of ROS, thereby contributing to the amelioration of cellular inflammation [254].

Expanding its functional spectrum, CypB emerges as an important regulator of adipogenesis [268,270]. Existing research has established a compelling link between ER stress and the dampening of adipogenic factors, ultimately leading to reduced fat accumulation [2]. Accumulating evidence in the literature has shown that ER stress contributes to the development and progression of obesity through multiple mechanisms [169]. This phenomenon has been primarily attributed to the inhibitory effects of CHOP on C/EBPβ, a critical regulator in the early stages of preadipocyte development [163]. However, recent investigations have revealed a paradigm shift, as CypB is now recognized for its role in downregulating CHOP expression [271]. Interestingly, the absence of CypB is associated with reduced lipid droplet formation in knockout cells, underscoring an enhanced adipogenic process under conditions of ER stress [268]. Most intriguingly, the alleviation of C/EBPβ repression acts as a catalyst, promoting the activation of C/EBPα and PPARγ, key transcription factors that exert maximal influence during the intermediate to late stages of cellular differentiation [36].

In summary, this comprehensive review holistically synthesizes the evolving understanding of the multifaceted roles of CypB in the intricate landscape of obesity-induced ER stress [268]. From its active participation in ER stress responses to its pivotal role in the regulation of adipogenesis, CypB’s significance reverberates across multiple physiological contexts. This nuanced exploration illuminates the intricate interplay between CypB and ER stress, extends its influence to mitigate inflammation and modulate adipogenesis, and provides a comprehensive view of its multifunctional capabilities.

## 6. The Cellular Consequences of Excessive ER Stress in the Adipose Tissue

### 6.1. Altered Lipid Metabolism and Dynamics of the Lipid Droplets

The ER plays a crucial role in lipid metabolism and homeostasis [272]. ER stress, caused by the accumulation of misfolded or unfolded proteins in the ER lumen, is a significant factor that affects lipid metabolism and droplet dynamics [273]. The disruption of lipid metabolism due to ER stress can have significant consequences for cellular lipid homeostasis and may contribute to the development of metabolic disorders [169,273,274,275].

ER stress can have an impact on lipid synthesis, including fatty acids and triglycerides. ER stress can upset the expression and activity of important enzymes, such as FAS and ACC, which are involved in lipogenesis [247]. This imbalance can lead to alterations in lipid synthesis and result in imbalances in the composition and species of lipid [5].

Furthermore, ER stress affects the dynamics and functioning of lipid droplets, which are intracellular organelles involved in lipid storage and metabolism [273]. Changes in lipid droplet-associated proteins, such as perilipins, adipose differentiation-related protein (ADRP), and seipin, induced by ER stress, can potentially impact lipid droplet formation, growth, and turnover, which, in turn, can alter cellular lipid storage capacity, lipolysis, and lipid utilization [276,277,278].

The interplay between ER stress and altered lipid metabolism contributes to lipotoxicity, a condition characterized by the accumulation of toxic lipid species, mitochondrial dysfunction, and cellular damage [274]. ER stress-induced lipotoxicity has been implicated in the pathogenesis of various metabolic disorders, including obesity, insulin resistance, NAFLD, and cardiovascular disease [131].

Understanding the molecular mechanisms underlying the crosstalk between ER stress and altered lipid metabolism is crucial for elucidating the pathogenesis of metabolic disorders and identifying potential therapeutic targets [169]. Modulating ER stress and restoring lipid homeostasis have emerged as potential strategies for mitigating the detrimental effects of altered lipid metabolism associated with ER stress.

In summary, ER stress disrupts lipid metabolism and alters lipid droplet dynamics, leading to imbalances in lipid synthesis, storage, and utilization [273]. These alterations contribute to lipotoxicity and the development of metabolic disorders [275]. Further research is needed to unravel the specific mechanisms underlying these changes and to explore therapeutic interventions that can restore lipid homeostasis and mitigate the adverse effects of ER stress on lipid metabolism [273].

### 6.2. Adipokine Dysregulation and Metabolic Inflammation

Adipocytes produce their own cytokines, also known as adipokines, causing chronic inflammation in the adipose tissue (AT) [279,280]. AT macrophages (ATM) intensify this metabolic dysfunction of adipocytes, increasing inflammation within the cells [281]. Leptin and adiponectin are major adipokines regulating lipid metabolism and glucose levels within the AT, and dysregulation of adipokines is associated with obesity [27]. A study also reported that the number of macrophages present in the AT is related to the actual adipocyte size [282,283].

The PERK pathway during excessive ER stress also activates cytokines such as TNF-a, IL-6, and IL-1β, a major contributor to the inflammation that induces obesity [4,284,285]. Increases in TNF-a and IL-6 cytokines show morphological changes in adipocytes, forming crown-like structures [79]. However, these cytokines are reported to reduce adipogenesis by inhibiting PPARγ and C/EBPα expression [147]. IL-6 production causes AT dysfunction that impairs differentiation of preadipocytes, and TNF-a alone is sufficient to inhibit the induction of PPARγ and C/EBPα [286].

In addition, activation of M1 macrophages causes proinflammatory effects in AT through secretion of IL-1B and TNF-a cytokines [287]. A study reported that adipocyte apoptosis accumulates macrophages and other immune cells around the dead cell, forming the crown-like shape [79,288]. The increase in crown-like structures in all fat depots has a positive relationship with obesity, and the change in shape causes the overall size of the adipocyte to increase, causing hypertrophic results [289,290]. M1 macrophages also induce insulin resistance, dysregulate AT homeostasis, and further exacerbate obese characteristics [291].

### 6.3. Correlation between Aged Adipose Tissue and ER Stress

Aging is associated with redistribution of adipose tissue in visceral organs [292]. It is recognized as a major source of chronic systemic inflammatory cytokines during aging (inflammaging) due to abundant inflammatory mediators and M1 pro-inflammatory macrophages [293]. Comprising various cell populations, adipose tissue includes not only adipocytes contributing to fat storage, but also non-adipocytes known as stromal vascular fraction (SVF), derived post-collagenase digestion, forming the extracellular matrix (ECM) and vasculature [294,295]. The majority of cells in the SVF are composed of white blood cells and adipose tissue stromal cells (ATSCs) [296]. Adipose tissue macrophages (ATM) are the major leukocyte population found in adipose tissue, presumed to promote inflammation in obesity and metabolic disorders [297].

Studies strongly suggest that ATSC, a constituent of internal visceral adipose tissue (including pre-adipocytes), is a primary cause of age-related adipose tissue inflammation [296]. Notably, elevated levels of TNFα in aged adipose tissue interfere with fat generation and correlate with increased expression of CHOP, a downstream target of the ER stress response pathway [298,299].

Furthermore, investigation of aged adipose tissue cells in mice revealed decreased autophagy, increased endoplasmic reticulum stress, and heightened inflammation [296]. The expression of the autophagy-related genes, autophagy related 7 (Atg7) and microtubule-associated protein 1A/1B-light chain 3-II (LC3-II) proteins decreases as levels of p62 and polyubiquitin accumulate, which coincides with decreased autophagy in aged rat kidneys [300,301]. Insufficient specific autophagy-related genes compromise cellular maintenance, notably impacting the lifespan of model organisms such as C. elegans and Drosophila, particularly under nutritional and oxidative stress [302]. Key lifespan-regulating pathways (Foxo3, SIRT1, mTOR, NF-κB, P53) modulate autophagy [303,304]. Activating autophagy through rapamycin highlights its potential in extending lifespan, emphasizing insights into aging mechanisms [305]. Notably, fibroblasts from long-lived mutant mice exhibit enhanced autophagy under stress conditions [306]. Autophagy-related genes significantly declined in aged adipose cells and worsened after stress induction [307]. New data now link heightened ER stress response and impaired autophagy alongside the accumulation of senescent cell progenitors to molecular events upstream of age-related adipose tissue inflammation [308,309].

Excessive weight in old age impacts physical function decline, loss of independence, and the development of frailty [308]. Delaying the aging process of adipose tissue is believed to prevent age-related diseases.

## 7. Conclusions

This review highlights the intricate relationship between ER stress and adipogenesis. The published findings in this review suggest that there is scientific evidence supporting the crosstalk between ER stress and adipogenesis. Their potential interplay is illustrated in Figure 2. Targeting ER stress pathways shows promise in treating adipogenesis-related disorders. Dysregulated adipogenesis contributes to obesity and metabolic disorders. Lifestyle changes such as exercise and diet affect ER stress and adipogenesis. Pharmacological options include agents that alleviate ER stress and modulate UPR pathways such as ATF6, IRE1 and PERK. Chemical chaperones and small molecule inhibitors have potential. Lifestyle interventions coupled with pathway modulation offer avenues for progress. However, research gaps remain in understanding mechanisms and clinical feasibility. Safety, efficacy and personalized approaches need to be explored. The link between ER stress and adipogenesis-related disorders provides opportunities for innovative interventions to improve metabolic health and patient outcomes.

## Figures and Tables

**Figure 1 nutrients-15-05082-f001:**
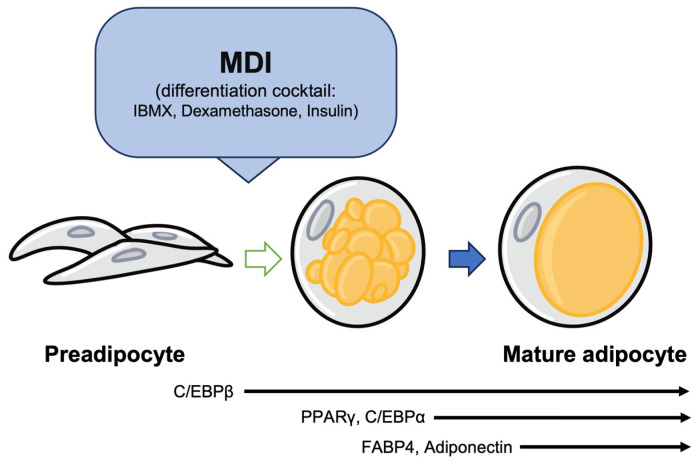
Adipogenesis factors involved in the stages of adipogenesis. C/EBPβ plays an important role in activating the expression of PPARγ and C/EBPα during the early stages of differentiation. PPARγ induces the expression of FABP4. Additionally, C/EBPα promotes the expression of adiponectin.

**Figure 2 nutrients-15-05082-f002:**
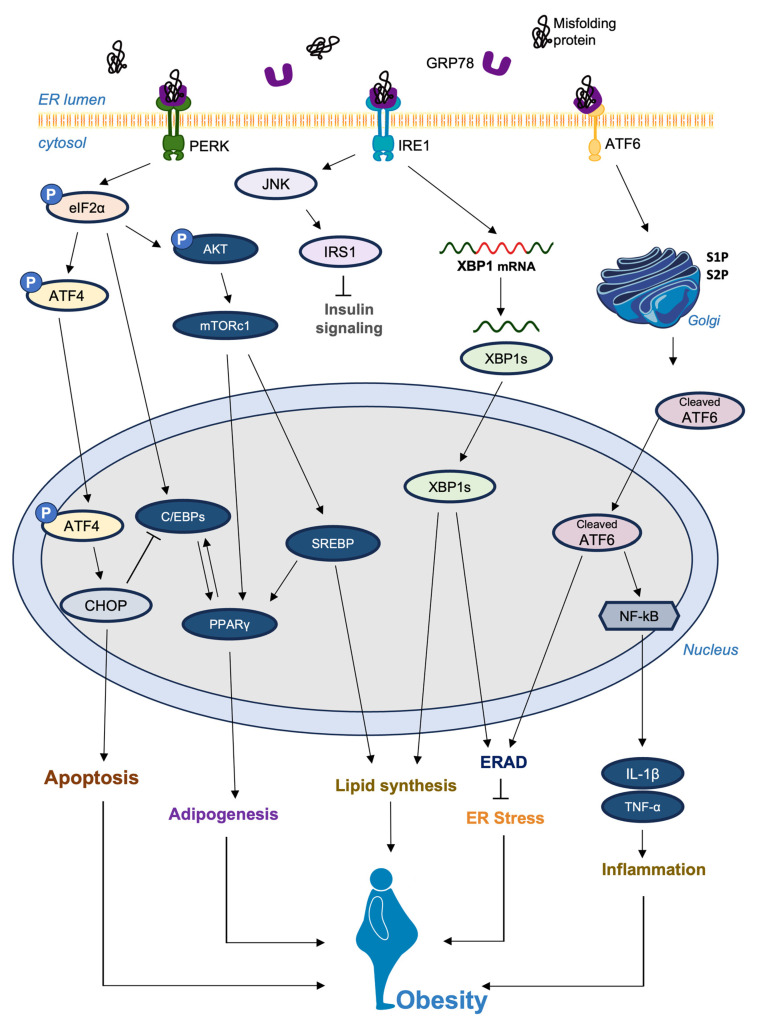
The potential association between the ER stress signaling pathway and obesity. ER stress induced by a variety of factors activates the unfolded protein response (UPR). PERK, IRE1α, and ATF6α, located in the ER membrane, act as UPR messengers and maintain ER stability by coordinating protein production and gene expression. PERK reduces protein synthesis by modifying eIF2α and induces CHOP through ATF4 mRNA translation. IRE1α splices XBP1 mRNA to generate XBP1 and induces genes linked to ER function. XBP1 enhances ER membrane formation, protein folding, transport, and ERAD. ATF6α is processed in the Golgi by S1P and S2P to release p50ATF6α (cleaved ATF6), which is pivotal for genes involved in ER protein folding and processing. All of these processes are multifactorial and highly interlinked.

**Table 1 nutrients-15-05082-t001:** XBP1s regulate various genes involved in lipid synthesis and lipid storage by promoting the expression of lipogenic genes. The table displays some of the key lipogenic genes promoted by XBP1s.

Modulators	Full Name	Roles in Lipogenesis
FAS	Fatty acid synthase	XBP1s promotes the expression of the FAS gene, contributing to fatty acid synthesis. FAS is an enzyme responsible for generating fatty acids and plays a crucial role in lipid metabolism.
SREBP1c	Sterol regulatory element-binding protein 1c	XBP1s regulates the expression of the SREBP1c gene, facilitating lipid synthesis. SREBP1c also activates other important genes related to lipid metabolism.
ACC	Acetyl-CoA carboxylase	XBP1s enhances the expression of the ACC gene, increasing the conversion of acetyl-CoA into fatty acids. This process is essential in fatty acid synthesis and is one of the key steps.
DGAT	Diacylglycerol O-Acyltransferase	XBP1s regulates the expression of the DGAT gene, promoting processes related to lipid droplets. This is associated with lipid storage
ChREBP	Carbohydrate-responsive element-binding protein	XBP1s controls the expression of the ChREBP gene, regulating the interaction between carbohydrate metabolism and fatty acid synthesis.
PLIN	Perilipin	XBP1s controls the expression of the PLIN gene, facilitating the perilipin protein found on the surface of lipid droplets. Perilipin stabilizes lipid droplets and regulates lipid storage and movement processes.
CIDE	Cell death-inducing DFFA-like effector	XBP1s contributes to the dynamics of lipid droplets by regulating the expression of certain genes within the CIDE gene family. These genes play a role in modulating the structure and function of lipid droplets.
ATGL	Adipose triglyceride lipase	XBP1s regulates the expression of the ATGL gene, controlling the breakdown of triglycerides in neutral fat. This process is associated with the movement of lipids within lipid droplets.
HSL	Hormone-sensitive lipase	XBP1s further regulates the breakdown of triglycerides in neutral fat by controlling the expression of the HSL gene. This process is related to energy metabolism.

## Data Availability

Not applicable.

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
