# Peer review of "Endoplasmic Reticulum Stress and Its Impact on Adipogenesis: Molecular Mechanisms Implicated"

_nutrients, 2023, doi:10.3390/nu15245082_

Round 1

Reviewer 1 Report

Comments and Suggestions for Authors

The review article titled "Endoplasmic Reticulum Stress and its Impact on Adipogenesis: Molecular Mechanisms Implicated" by Gyuhui Kim et al. explores the role of dysregulation in the unfolded protein response (UPR) pathways, namely IRE1-XBP1 and PERK signaling, in regulating adipocyte differentiation, lipid metabolism, and tissue inflammation. It specifically explores the effects of endoplasmic reticulum (ER) stress on the regulation of adipogenic transcription factors such as Proliferator-Activated Receptor γ (PPARγ) and CCAAT-Enhancer-Binding Proteins (C/EBPs). Understanding the mechanisms that correlate ER stress and adipogenesis is crucial for developing therapeutic agents in metabolic disorders.

The review comprises a concise introduction with clear background information. The authors have gathered a significant amount of information from various peer-reviewed research articles and generated a review article that highlights the importance of ER stress in adipocyte biology. Overall, the review effectively summarizes key findings and should be of interest to adipocyte biologists.

Minor issues:

  1. The abstract does not mention the role of ATF6, one of the three main transmembrane proteins in the UPR pathway.

  1. Although the abstract refers to the discussion of potential therapeutic strategies targeting ER stress in the main article, it is not actually discussed.

  1. Sections 4.3 and 4.3.1 appear to be repetitive.

Additionally, please check for spelling errors, such as "River X receptor" for "Liver X receptor" on page #11.

Comments on the Quality of English Language

Minor revision.

Author Response

Response to reviewer’s comments

Reviewer1:

The review article titled "Endoplasmic Reticulum Stress and its Impact on Adipogenesis: Molecular Mechanisms Implicated" by Gyuhui Kim et al. explores the role of dysregulation in the unfolded protein response (UPR) pathways, namely IRE1-XBP1 and PERK signaling, in regulating adipocyte differentiation, lipid metabolism, and tissue inflammation. It specifically explores the effects of endoplasmic reticulum (ER) stress on the regulation of adipogenic transcription factors such as Proliferator-Activated Receptor γ (PPARγ) and CCAAT-Enhancer-Binding Proteins (C/EBPs). Understanding the mechanisms that correlate ER stress and adipogenesis is crucial for developing therapeutic agents in metabolic disorders.

The review comprises a concise introduction with clear background information. The authors have gathered a significant amount of information from various peer-reviewed research articles and generated a review article that highlights the importance of ER stress in adipocyte biology. Overall, the review effectively summarizes key findings and should be of interest to adipocyte biologists.

Minor issues:

  1. The abstract does not mention the role of ATF6, one of the three main transmembrane proteins in the UPR pathway.

Answer) Thank you for pointing this out. As suggested by the reviewer, we have mentioned the role of ATF6 in the abstract and introduction.

Please see line 17, and 44.

  1. Although the abstract refers to the discussion of potential therapeutic strategies targeting ER stress in the main article, it is not actually discussed.

Answer) We apologize for this error. The manuscript focuses on the mechanisms associated with ER stress and adipogenesis rather than therapeutic strategies, although some therapeutic strategies were mentioned in each section. As suggested by the reviewer, we have changed the sentence in the abstract from “investigate potential therapeutic strategies that target ER stress with an aim to mitigate obesity-related disorders” to “discuss the potential roles of the molecular chaperones, cyclophilin A and Cyclophilin B in adipogenesis”.

Please see line 22-23

  1. Sections 4.3 and 4.3.1 appear to be repetitive.

Answer) Thank you for your critical comment.  However, Section 4.3 serves as an introduction to the interplay between ER stress and adipogenesis mechanisms. Subordinate to Section 4.3, Section 4.3.1 elucidates the correlation between the UPR pathway and the adipogenesis mechanisms. From Section 4.3.2 to section 4.3.5, the relationship between er stress and other mechanisms in adipogenesis is explained.

  1. Additionally, please check for spelling errors, such as "River X receptor" for "Liver X receptor" on page #11.

Answer) We apologize for this mistake. In accordance with the reviewer’s suggestion, we corrected the spelling error.

Please see line 472.

Reviewer 2 Report

Comments and Suggestions for Authors

The authors of this review aimed to scrutinize the crosstalk and its underpinned molecular mechanisms between ER stress and generation/maturation of adipocytes, as well as to trace the connection between ER stress and adipogenesis in a framework of adipogenesis-associated metabolic disorders, mainly obesity.
The authors did a lot of work, processed and presented a large amount of scientific data, however, there are serious shortcomings that need to be corrected for the review.
1.       The main idea of the review is not fully argumentized.    There is no clear definition of ER-stress.    ER-stress and UPR are sometimes considered separately as unrelated subjects, and the causal relationship is compromised.    The position of authors about ER-stress and obesity is blurred – is it obese environment that causes ER-stress or ER-stress triggers obesity (lines 31, 125-128, 173-175, 302-304, 496-498).    The narrative runs often as if in parallel – the facts about ER-stress and adipogenesis in some cases have little links.

2.       The review is not good structured, hard-to-read text, there is a quite amount of text repeats.    Sometimes it seems that almost each new section was written by another person, reiterating already mentioned theory about UPR components/ UPR signaling pathways/ adipogenesis (example: lines 96-98, 167-172, 179-185).    Some sections are better to be combined into one, for example parts about insulin resistance (3.2 and 4.3.1).

3.       The review length is large though some aspects are outlined superficially or overlooked by the authors.    As for example:
•    there are no solid proofs that mesenchymal stem cells serve as progenitors of adipocytes inside the organism (it is even discussed in the review authors refer to (Cawthorn et al., 2012))
•    Having discussed adipogenesis and lipid accumulation so extensively, there is no, even short, information about mechanisms of lipid synthesis.
•    Section 5.1 – no mechanisms described
•    Therapeutic strategies are mentioned in abstract but little attention was paid to it in the text.

4.       One of the key disadvantages – the review lacks studies discussing pleiotropic effect of ER-stress – not only as pathological (as it was introduced by the authors) but as physiological process that may contribute to cell differentiation.    The physiological role of ER-stress is discussed only for IRE1-XBP1 signaling pathway, however there are studies postulating that the functioning of all three UPR activation pathways is required for successful adipogenesis (Sha et al., 2009, Cell Metab.;    Bobrovnikova-Marjon E et al., 2012, Mol. Cell.    Biol.;    Lowe et al., 2012, Int.    J. Obesity, Turishcheva et al., 2022, Biochemistry (Mosc)).

5.       The text contains 284 references, however, those which were published during the last 5 years amounts to less than a third.

To conclude, despite the fact that the review contains several well-written sections (Sec. 1, 2, 6), it, as a whole, requires a serious improvement and more logical structure.

Author Response

November 30, 2023

Manuscript ID: nutrients-2714484
Type of manuscript: Review
Title: Endoplasmic Reticulum Stress and its Impact on Adipogenesis: Molecular
Mechanisms Implicated
Authors: Gyuhui Kim, Jiyoon Lee, Joohun Ha, Insug Kang, Wonchae Choe *

Dear Editor-in-Chief of Nutrients,

We would like to thank you for the letter dated November/21/2023, and the opportunity to resubmit a revised version of this manuscript. We would also like to take this opportunity to express our thanks to the reviewers for the positive feedback and helpful comments for correction or modification.

We believe the opinions of the reviewers have resulted in an improved revised manuscript, which you will find uploaded alongside this document. The manuscript has been revised to address the reviewers’ comments, which are appended alongside our responses to this letter. The revision has been developed in consultation with all coauthors, and each author has given approval to the final form of this revision.

We very much hope the revised manuscript is accepted for publication in Nutrients.

Sincerely yours,

Wonchae Choe, PhD

Department of Biochemistry and Molecular Biology School of Medicine, Kyung Hee University

#26, Kyungheedae-ro, Dongdaemun-gu Seoul, 02447, Republic of Korea

Tel: +82-2-961-0940

Fax: +82-2-959-8168

Reviewer 2

the authors of this review aimed to scrutinize the crosstalk and its underpinned molecular mechanisms between ER stress and generation/maturation of adipocytes, as well as to trace the connection between ER stress and adipogenesis in a framework of adipogenesis-associated metabolic disorders, mainly obesity.
The authors did a lot of work, processed and presented a large amount of scientific data, however, there are serious shortcomings that need to be corrected for the review.

1-1.  The main idea of the review is not fully argumentized. There is no clear definition of ER-stress.    ER-stress and UPR are sometimes considered separately as unrelated subjects, and the causal relationship is compromised. 

Answer) Thank you for your insightful comment. As suggested by the reviewer, we have included a clear definition of ER stress. We also briefly discussed the relationship between ER stress and UPR.

Please see line: 34-40, 41-42.  

1-2. The position of authors about ER-stress and obesity is blurred – is it obese environment that causes ER-stress or ER-stress triggers obesity (lines 31, 125-128, 173-175, 302-304, 496-498).  The narrative runs often as if in parallel – the facts about ER-stress and adipogenesis in some cases have little links.

Answer) Thank you for your critical comment, again. We added the sentence that ER stress and obesity have a bidirectional interaction. Please see line: 48-54

2.   The review is not good structured, hard-to-read text, there is a quite amount of text repeats.    Sometimes it seems that almost each new section was written by another person, reiterating already mentioned theory about UPR components/ UPR signaling pathways/ adipogenesis (example: lines 96-98, 167-172, 179-185). 

 Answer) We apologize for this error. As suggested by the reviewer, the repetitive parts about UPR components/ UPR pathways/ adipogenesis have been deleted.

Please see line 95-102, 172-173, 192-199, 206-212 242-243 287-289.

2-1. Some sections are better to be combined into one, for example parts about insulin resistance (3.2 and 4.3.1).

Answer) In accordance with the suggestion of the reviewer, the parts related to insulin resistance have been combined into a single part. Please see line 432-441

3.       The review length is large though some aspects are outlined superficially or overlooked by the authors.    As for example:
3-1  there are no solid proofs that mesenchymal stem cells serve as progenitors of adipocytes inside the organism (it is even discussed in the review authors refer to (Cawthorn et al., 2012))
Answer) Thanks for your critical comments. As suggested by the reviewer, we corrected the sentence about mesenchymal stem cells. Please see line 77-79

3-2   Having discussed adipogenesis and lipid accumulation so extensively, there is no, even short, information about mechanisms of lipid synthesis.
Answer) As suggested by the reviewer, we have added brief information on the mechanism of lipid synthesis. Please see line: 166-172.

3-3   Section 5.1 – no mechanisms described.

Answer) We apologize for this mistake. In accordance with the reviewer's suggestion, we have described the mechanism. Please see line 554-558.

3-4    Therapeutic strategies are mentioned in abstract but little attention was paid to it in the text.

Answer) We apologize for this error. The manuscript focuses on the mechanisms associated with ER stress and adipogenesis rather than therapeutic strategies, although some therapeutic strategies were mentioned in each section. As suggested by the reviewer, we have changed the sentence in the abstract from “investigate potential therapeutic strategies that target ER stress with an aim to mitigate obesity-related disorders” to “discuss the potential roles of the molecular chaperones, cyclophilin A and Cyclophilin B in adipogenesis”.

Please see line 22-23

4.  One of the key disadvantages – the review lacks studies discussing pleiotropic effect of ER-stress – not only as pathological (as it was introduced by the authors) but as physiological process that may contribute to cell differentiation.    The physiological role of ER-stress is discussed only for IRE1-XBP1 signaling pathway, however there are studies postulating that the functioning of all three UPR activation pathways is required for successful adipogenesis (Sha et al., 2009, Cell Metab.;    Bobrovnikova-Marjon E et al., 2012, Mol. Cell.    Biol.;    Lowe et al., 2012, Int.    J. Obesity, Turishcheva et al., 2022, Biochemistry (Mosc)).

Answer) Thank you for your critical comment. As suggested by the reviewer, the additional roles of ER stress under physiological conditions such as differentiation, and aging have been added to the manuscript. We also added the sentences that the ER stress pathways (IRE1, PERK, ATF6) collectively drive both lipogenesis and adipogenesis, and that inhibiting all of them is pivotal to curb lipogenesis and delay the onset of obesity.

Please see line 199-203, 224-241, 248-257, 272-280, 657-691.

  1. The text contains 284 references, however, those which were published during the last 5 years amounts to less than a third.

 Answer) We apologize for this mistake. In accordance with the reviewer's suggestion, we have updated the manuscript with the latest reference papers. Please see the reference papers.

  1. To conclude, despite the fact that the review contains several well-written sections (Sec. 1, 2, 6), it, as a whole, requires a serious improvement and more logical structure.

Answer) As suggested by the reviewer, we have revised the manuscript. We hope that the reviewer will be satisfied with it.

Reviewer 3 Report

Comments and Suggestions for Authors

I only have a few minor comments to raise with the authors to improve the quality of the work:

- include a table with the acronyms of all the genes and proteins in the review;

- in paragraph 2 the authors are requested to emphasize more strongly the dysregulation of the adipogenesis process in obesity;

- Figure captions should be more descriptive, and not just include the title.

- I would advise the authors to add a paragraph referring to the correlation between ER stress and ageing of adipose tissue ( https://www.ncbi.nlm.nih.gov/pmc/articles/PMC5115904/; https://pubmed.ncbi.nlm.nih.gov/37511435/)

Author Response

November 30, 2023

Manuscript ID: nutrients-2714484
Type of manuscript: Review
Title: Endoplasmic Reticulum Stress and its Impact on Adipogenesis: Molecular
Mechanisms Implicated
Authors: Gyuhui Kim, Jiyoon Lee, Joohun Ha, Insug Kang, Wonchae Choe *

Dear Editor-in-Chief of Nutrients,

We would like to thank you for the letter dated November/21/2023, and the opportunity to resubmit a revised version of this manuscript. We would also like to take this opportunity to express our thanks to the reviewers for the positive feedback and helpful comments for correction or modification.

We believe the opinions of the reviewers have resulted in an improved revised manuscript, which you will find uploaded alongside this document. The manuscript has been revised to address the reviewers’ comments, which are appended alongside our responses to this letter. The revision has been developed in consultation with all coauthors, and each author has given approval to the final form of this revision.

We very much hope the revised manuscript is accepted for publication in Nutrients.

Sincerely yours,

Wonchae Choe, PhD

Department of Biochemistry and Molecular Biology School of Medicine, Kyung Hee University

#26, Kyungheedae-ro, Dongdaemun-gu Seoul, 02447, Republic of Korea

Tel: +82-2-961-0940

Fax: +82-2-959-8168

Reviewer 3

I only have a few minor comments to raise with the authors to improve the quality of the work:

  1. include a table with the acronyms of all the genes and proteins in the review;

 Answer) We appreciate the reviewer's suggestion. A table containing acronyms for all the genes and proteins mentioned in the review has been added at line 719.

  1. in paragraph 2 the authors are requested to emphasize more strongly the dysregulation of the adipogenesis process in obesity.

 Answer) As suggested by the reviewer, we have changed the sentence from "dysregulation of the adipogenesis process in obesity" to "the process of adipogenesis in obesity".

Please see line 62.

  1. Figure captions should be more descriptive, and not just include the title.

 Answer) As per the reviewer's suggestion, we have made the correction in Figure captions.

Please see line 149-157, 329-332.

  1. I would advise the authors to add a paragraph referring to the correlation between ER stress and ageing of adipose tissue

Answer). We appreciate the reviewer's suggestion. New section 6.3 includes the correlation between ER stress and the ageing of adipose tissue.

Please see line 21, 657-692.
 (https://www.ncbi.nlm.nih.gov/pmc/articles/PMC5115904/; https://pubmed.ncbi.nlm.nih.gov/37511435/)

Round 2

Reviewer 2 Report

Comments and Suggestions for Authors

All necessary changes in the manuscript were made according to my comments.